Assessing the ecological risk of heavy metal sediment contamination from Port Everglades Florida USA

Giarikos Dimitrios G. giarikos@nova.edu 1 2
White Laura 3
Daniels Andre M. 4
Santos Radleigh G. 2 5
Baldauf Paul E. 2 3
Hirons Amy C. 2 3
1 Chemistry and Physics, Nova Southeastern University , Fort Lauderdale , FL , United States of America
2 SECLER: Study of Environmental Conservation through Leading-Edge Research, Nova Southeastern University , Fort Lauderdale , FL , United States of America
3 Department of Marine and Environmental Sciences, Nova Southeastern Univeristy , Fort Lauderdale , FL , United States of America
4 Wetland and Aquatic Research Center, U.S. Geological Survey , Davie , FL , United States of America
5 Department of Mathematics, Nova Southeastern University , Fort Lauderdale , FL , United States of America
Latimer James
Electronic publication date: 2023 Nov 14
Publication date: 2023
Volume: 11
Electronic Location ID: e16152
Received 2023 Apr 19; Accepted 2023 Aug 31
Copyright: ©2023 Giarikos et al.
Copyright year: 2023
Copyright holder: Giarikos et al.
License: This is an open access article distributed under the terms of the Creative Commons Attribution License, which permits unrestricted use, distribution, reproduction and adaptation in any medium and for any purpose provided that it is properly attributed. For attribution, the original author(s), title, publication source (PeerJ) and either DOI or URL of the article must be cited.
License URL: https://creativecommons.org/licenses/by/4.0/

Keywords: Heavy metals, Arsenic, Port Everglades, Ecological risks, Contamination, Sediment, Sediment cores, Molybdenum, Florida, Coral reef

Funding: The President’s Faculty Research and Development Grant (PFRDG) index number 334808 at Nova Southeastern University (NSU), Florida NSU’s consortium SECLER (Study of Environmental Conservation through Leading-Edge Research) U.S. Geological Survey, Wetland and Aquatic Research Center, Davie, Florida This work was financially supported by the President’s Faculty Research and Development Grant (PFRDG) index number 334808 at Nova Southeastern University (NSU), Florida, and NSU’s consortium SECLER (Study of Environmental Conservation through Leading-Edge Research). Florida. In-kind support was provided by U.S. Geological Survey, Wetland and Aquatic Research Center, Davie, Florida (Andre Daniels) and U.S. Geological Survey, Coastal and Marine Science Center, St. Petersburg, Florida (Kyle Kelso and Nancy DeWitt). The funders had no role in study design, data collection and analysis, decision to publish, or preparation of the manuscript.

==============================
Port sediments are often contaminated with metals and organic compounds from anthropogenic sources. Remobilization of sediment during a planned expansion of Port Everglades near Fort Lauderdale, Florida (USA) has the potential to harm adjacent benthic communities, including coral reefs. Twelve sediment cores were collected from four Port Everglades sites and a control site; surface sediment was collected at two nearby coral reef sites. Sediment cores, sampled every 5 cm, were analyzed for 14 heavy metals using inductively coupled plasma-mass spectrometry. Results for all three locations yielded concentration ranges (µg/g): As (0.607–223), Cd (n/d–0.916), Cr (0.155–56.8), Co (0.0238–7.40), Cu (0.004–215), Pb (0.0169–73.8), Mn (1.61–204), Hg (n/d–0.736), Mn (1.61–204), Ni (0.232–29.3), Se (n/d–4.79), Sn (n/d–140), V (0.160–176), and Zn (0.112–603), where n/d = non-detected. The geo-accumulation index shows moderate-to-strong contamination of As and Mo in port sediments, and potential ecological risk indicates moderate-to-significantly high overall metal contamination. All four port sites have sediment core subsamples with As concentrations above both threshold effect level (TEL, 7.24 µg/g) and probable effect level (PEL, 41.6 µg/g), while Mo geometric mean concentrations exceed the background continental crust level (1.5 µg/g) threshold. Control site sediments exceed TEL for As, while the reef sites has low to no overall heavy metal contamination. Results of this study indicate there is a moderate to high overall ecological risk from remobilized sediment due to metal contamination. Due to an imminent dredging at Port Everglades, this could have the potential to harm the threatened adjacent coral communities and surrounding protected habitats.

Introduction

Coastal sediments associated with commercial activities can be laden with inorganic and organic pollutants (Benlahcen et al., 1997; USACE, 2004; Qian et al., 2015; Krek et al., 2018; Armiento et al., 2020). In this paper, sediment will refer to any organic or inorganic particle, natural or human-made, that has settled to the bottom of a body of water (Power & Chapman, 2018). Contaminants including bacteria (Droppo, 2001), metals present as metallic cations (Salomons et al., 1987), and nonpolar contaminants, such as solvents or many pesticides (El-Nahhal & El-Nahhal, 2021), may be sorbed to sediment surfaces or to organic materials (biofilms) that coat those particles (Headley et al., 1998). The chemistry of these reactions is complex, but it is possible for sediment contaminants to remobilize through desorption (Du Laing et al., 2009). These remobilized contaminants can be extremely useful indicators for monitoring anthropogenic impacts within aquatic environments (Ergin et al., 1991; Balls et al., 1997; Atgin et al., 2000; Rifkin, Gwinn & Bouwer, 2004; USACE, 2004). Contaminated sediments in aquatic ecosystems throughout the world have been linked with potential human and ecological risks (Long et al., 1996; Turgeon et al., 1998; Rifkin, Gwinn & Bouwer, 2004; USACE, 2004). When dredged from waterways, sediments are often contaminated with historically derived chemical pollutants. These contaminants may be distributed, ingested, or absorbed by marine organisms which could result in toxic effects or bioaccumulation in the food web (USEPA, 2021).

Sediment resuspension and movement is common in semi-enclosed, tidal influenced systems such as ports. During this process, heavy metals can be released (remobilized) into the water column through desorption and pose a potential risk to biological organisms, particularly in benthic communities during dredging (Mudroch, 1983; Filho et al., 2004; USEPA, 2021; Wei et al., 2021). The known desorption rates of metals, especially during port dredging events, are limited and potentially variable, dependent upon multiple factors. Clay, for example, is unlikely to absorb heavy metals, while dissolved organic matter and phosphorus based compounds favor heavy metal adsorption (Miranda et al., 2022). Abiotic variables such as pH and salinity can dictate the separation of heavy metals between the water, suspended particulate matter, and settled sediment (Feng et al., 2017). Evidence of heavy metal remobilization from sediment into the water column has been displayed in the sentinel crab (Macrophthalmus spp.), common periwinkle (Littorina littorea), and Sydney rock oyster (Saccostrea glomerata) (Davies & Uyi, 2009; Hedge, Knott & Johnston, 2009; Saadati et al., 2020). Partial mortality of nearby corals has also been attributed to significant sediment accumulation that resulted from the 2013–2015 dredging of the Port of Miami (Miller et al., 2016). Resuspension of dredged sediments poses an immediate physical hazard to benthic communities through turbidity, but it can also have a chemical impact, as contaminants can be resuspended in the water column and bioabsorbed. Entrainment of suspended sediment during a dredging project is difficult to maintain and environmentally sensitive coral reef communities could be at risk. Eggs and larvae of both invertebrate and vertebrate species are at the highest risk from sedimentation, so environmental windows are used (when dredging pauses) to limit resuspension during spawning (Todd et al., 2015).

Port Everglades, located in direct proximity to both the US Intracoastal Waterway and nearshore coral reef tracts, exhibits characteristics of an estuarine environment, as it is semi-enclosed and connects to the ocean. Estuarine waters, some of the most productive in the world, are subject to significant anthropogenic impacts (Underwood & Kromkamp, 1999; Chapman & Wang, 2001). Historic contamination, such as decades’ worth of anthropogenic port contaminants, is a significant environmental concern which can have harmful effects to both benthic and water column species (Barnes, 1968; Varanasi et al., 1985; Nichols et al., 1986; French, 1993; Vallette-Silver, 1993; Stein et al., 1995; Virkanen, 1998).

The planned expansion of the commercial Port Everglades includes blasting and dredging of bottom sediment to accommodate larger vessels (USACE, 2022). Bach, Nielsen & Bollwerk (2017) showed that underwater blasting and dredging of bedrock during a quay construction in Sisimiut, West-Greenland in 2012 dispersed fine grained and/or organic sediment carrying heavy metals greater than 350 m from the construction site. Analyses revealed that bioavailable heavy metals (As, Cd, and Pb) from the dispersed sediment accumulated in mussels at intervals of 150, 250, and 350 m from the quay. The ecological risk posed by the resuspension of contaminated sediment is great to the adjacent coral reef threatened species (Miller et al., 2016; Tuttle & Donahue, 2022; FDEP, 2023a; USEPA, 2023), but also to other habitats and communities in the vicinity of the port such as mangroves and seagrass beds which support not only invertebrate species but also nektonic organisms such as birds (wading birds), reptiles (sea turtles), fish, and marine mammals (manatees), many of which are federally protected species (FMNH, South Florida Aquatic Environments, 2020; FDEP, 2023b; FDEP, 2023c; Miamiscapes, 2023).

The goal of this study is to determine the concentrations of heavy metals in the port sediment and evaluate the environmental risk to the nearby coral reef community due to the remobilization of contaminated port sediments by calculating ecological risk indices (geo-accumulation, pollution load, and potential ecological risk) and comparing the concentrations to the known TEL and PEL values. In this study, we define heavy metals as trace metals and metalloids (As and Se) that have a high atomic weight and a density that is at least five times that of water. Fourteen heavy metals (As, Cd, Co, Cr, Cu, Pb, Hg, Mn, Mo, Ni, Se, Sn, V, and Zn) were analyzed and quantified to compare concentrations among three locations (port, control, and coral reef). The data reveal high concentrations of heavy metals that will likely be resuspended and distributed to the nearby coral reef tract during any blasting, dredging, and disposal operation of sediment material. This research lays the groundwork for future anthropogenic contamination research and environmental toxicology studies on habitats and communities near ports. The findings can be used as a comparison for future port dredging impact studies and aid in developing mitigation strategies.

Materials & Methods

Permits for sediment core collection were provided by Broward County (Permit #ES2019-11), US Army Corps of Engineers (Permit #SAJ-2019-01644(NW-LCK)), and Florida Department of Environmental Protection (Permit #05131935, Project #06-375933-001-EE). Twelve, 1–2 m sediment cores were collected July 9–11, 2019. Ten sediment cores were taken from four Port Everglades sites and two cores from a control location, West Lake (WL), approximately three miles south of Port Everglades along the US Intracoastal Waterway (ICW). West Lake was selected as a control since it is in West Lake Park, a 1500-acre mangrove preserve where no motorized vessels have been permitted since 1988, all of which greatly reduced anthropogenic input and port influence (SFLSS, 1988; FDEP, 2021; Miller Legg Consulting, 2022). Three cores were taken from Dania Cutoff Canal (DCC), three from Park Education Center (PEC), two from Park Headquarters (PHQ), and two from South Turning Basin (STB). Two cores were collected from PHQ and STB due to boat time availability and difficulty collecting these cores. All sites are in the vicinity of ship traffic, vessels, equipment maintenance, and fueling activities. All cores were collected in water depths less than 2 m along the water body margin representing the main port environment. In addition, divers collected three surface sediment samples in 55 oz sterilized sampling bags from the seafloor (4–5 m in depth), at two reef sites (RF) 1.5 km north and south of the inlet to Port Everglades (Fig. 1). Replicate sediment cores within each site were taken to compare 14 heavy metal (As, Cd, Co, Cr, Cu, Pb, Hg, Mn, Mo, Ni, Se, Sn, V, and Zn) concentration variations within each site, as marine sediments are subject to dispersal by water movement (Wang & Andutta, 2012).

Figure 1 Map of Port Everglades, Florida (USA) showing the Intracoastal Waterway, Port, the core, and grab sampling locations.

Park Headquarters (PHQ), Park Education Center (PEC), South Turning Basin (STB), Dania Cutoff Canal (DCC), West Lake (WL), South Coral Reef (SR), North Coral Reef (NR), and the three coral reef tracts. The map was created using ArcGIS® software (Esri, Redlands, CA, USA). The data source of the reef map was obtained from the Florida Fish and Wildlife Conservation Commission-Fish and Wildlife Research Institute.

Sediment cores were collected as previously described in White (2021). Specifically, a Bradford pneumatic vibrator with an air compressor was clamped to a 7.5 cm diameter and 3 m length aluminum barrel supported by a vibracore extraction land-based tripod. Core catchers were added at the base of each barrel to ensure complete recovery of the sediments, and the barrels were cut to length and capped. The cores were split longitudinally at the US Geological Survey (USGS) St. Petersburg Coastal and Marine Science Center using a custom-made sediment core splitter. Each split core was wrapped in plastic and placed inside plastic sleeves to prevent desiccation. Each core was stored horizontally at 4 °C in a Continental CH3R-GD refrigerator. High resolution photos were taken of each split core prior to sampling (Fig. S1).

Digestion and analysis

Subsamples of sediment were taken as previously described in White (2021). Specifically, subsamples of sediment (1 cm3, ∼2.0 g) at 5 cm intervals were taken along the entire length of each core from surface to base. Each sample was washed three times with ultrapure deionized water (18.2 M Ω) from a Barnstead water purification system. The sediments were pre-dried overnight in a VWR drying oven for 18 h at 80 °C, followed by 5 h in a Fischer Scientific isotemp vacuum oven (model 282A) at 80 °C at a pressure below 10−2 torr, equipped with a Welch 1400 DuoSeal vacuum pump and a liquid nitrogen trap that resulted in completely dried samples. The dry weight of each sample was recorded and US Environmental Protection Agency (EPA) digestion method 3050B was used (Environmental Protection Agency (EPA), 1996).

Heavy metal analyses were performed as the University of Southern Mississippi’s Center for Trace Analysis as previously described in White (2021) by using a sector-field inductively coupled plasma mass spectrometer (ICP-MS) (ThermoFisher Element XR) with a Peltier-cooler spray chamber (PC-3; Elemental Scientific, Inc.). Prior to analysis, digested samples were diluted 5-fold in 0.64 M ultrapure nitric acid (Seastar Baseline) containing 2 ppb indium as an internal standard. Diluted samples were held in acid-washed Teflon autosampler vials. Mass spectrometer scans were performed in low (Cd-111, Hg-199,200,201,202, Pb-208), medium (Al-27, V-51, Cr-52, Mn-55, Fe-56, Co-59, Ni-60, Cu-63, Zn-66), and high (As-75, Se-77,82) resolution, depending on the isotope. Mo-98 was monitored to correct for MoO+ interference on Cd. Standardization was by use of external standards, with a high standard and a blank re-run every eight samples. For the elements (Hg, Se) where multiple isotopes were determined, no significant analytical differences were noted between the isotopes. Two USGS reference water concentrations were also assessed as part of each analytical run to verify the standardization. In several cases, sample calibration was also verified by standard additions. Blanks of ultrapure deionized water, hydrogen peroxide, and trace metal basis nitric acid (10%) were used for quality control purposes. Detection limits were calculated as three times the standard deviation of the blank (Table S1). Standard reference material SRM 2702-Inorganics in Marine Sediment (MilliporeSigma) was used to evaluate reliability of the analytical method and calculate percent recoveries (Table S2).

West Lake control site and continental crust composition

A control site for comparison with the port sites was established at West Lake (WL), located approximately 3 miles south of the port, outside of its influence. This site is part of West Lake Park, which includes a three-mile strip of mangrove estuary and uplands where no motorized boats are permitted. Since comprehensive background values for marine sediments for all 14 heavy metals have not been determined for this area, the port and reef heavy metal concentrations were also compared to the two cores that were collected from WL.

The elemental composition of the Earth’s upper crust, also referred to as background elemental composition concentrations, is used as a tool for assessing geochemical anomalies (heavy metal contamination) (Table S3). Since comprehensive background values for marine sediments for all 14 heavy metals have not been determined for this area, the concentrations were compared to continental crust values derived from the post-Archean Australian average shale (PAAS), European shale composite (ES), and North American shale composite (NASC) (Taylor & McLennan, 1995; Al-Mutairi & Yap, 2021). The sediment cores collected were at the coastal margin on continental crust. The transition between continental crust and oceanic crust is several miles from shore and any oceanic crust weathering is highly unlikely to impact the coastal sites (Webb, 0000; Britannica, 2023).

Geo-accumulation index

The geo-accumulation index (Igeo) is a quantitative measure of the degree of contamination in sediments (Förstner, 1980) and was used here to measure the pollution intensity at individual sample locations. The Igeo is calculated as follows: Igeo= log2Cn1.5xBn.

Cn = concentrations within the sediment cores and Bn = background continental crust levels. Rudnick & Gao (2014) provide values that quantify the degree of contamination: 4-5: strongly to extremely contaminated; 3-4: strongly contaminated; 2-3: moderately to strongly contaminated; 1-2: moderately contaminated; 0-1: uncontaminated to moderately contaminated; <0: uncontaminated.

Pollution load index

The pollution load index (PLI) was developed by Tomlinson et al. (1980) and is calculated using contamination factors (CF), represented by the concentrations of the sample metals and the background continental crust values Cmetal/Cbackground. The calculation for PLI is: PLI=CF1xCF2xCF3x…CFn1/n

where n = number of elements. This approach assesses total contamination load within the sediment and provides a PLI value that explains overall metal pollution within each sample. A sample with a PLI>1 is classified as polluted while a sample with a PLI<1 indicates no contamination (Tomlinson et al., 1980; Ray et al., 2006; Badr et al., 2009).

Threshold effect levels (TEL) and Probable effect levels (PEL)

The Florida Department of Environmental Protection created sediment quality assessment guidelines (SQAGs) to address coastal ecosystem contamination concerns. Numerical SQAGs were derived for nine metals (As, Cd, Cr, Cu, Hg, Ni, Pb, tributyltin, Zn) that occur in Florida coastal sediments. A threshold effect level (TEL) and probable effect level (PEL) were developed for these metals as powerful tools to assess contaminant levels in sediment (Macdonald et al., 1996). The TEL is the concentration below which adverse effects rarely occur to benthic organisms, while the PEL is the concentration above which adverse effects frequently occur (Geoenvironmental Engineering, 2015; Thompson & Wasserman, 2015).

Potential ecological risk index

The potential ecological risk index (PER) determines the degree of contamination for combined metal concentrations within each sediment sample (Guo, Xianbin & Zhanguang, 2010). The PER is calculated as: PER= ∑E

E=TC

C=Ca/Cb

where Ca = element content within sample, Cb = reference value of the element, and T = toxic response factor for metals: Mn and Zn =1, Cr =2, Cu and Pb =5, Ni =6, As =10, and Cd =30 (Hakanson, 1980; Fu et al., 2009; Guo, Xianbin & Zhanguang, 2010; Cao, Hong & Liu, 2015).

Ecological indices

Every ecological index has limitations as each assesses different heavy metal pollutants of importance. Geo-accumulation index (Igeo), potential ecological risk (PER), pollution load index (PLI), and enrichment factor (EF) are widely used for the assessment of the degree of pollution and health status of sediment. The advantage of the Igeo and EF indices is that they can identify whether the level of a specific heavy metal in sediment is due to anthropogenic or natural input or a combination of both (Schropp et al., 1990; Shafie et al., 2013; Zahra et al., 2014; Abdullah, Sah & Haris, 2020). The PLI identifies overall/total heavy metal contamination, and the PER identifies degrees of total metal contamination (Jahan & Strezov, 2018; Ali et al., 2022; Ranjani et al., 2021).

Concentrations of Fe and Al were not assessed in this study so EF values could not be determined. These two metals are most used for normalization to calculate the enrichment factor (EF) since normalization is used to differentiate if the metal source is of anthropogenic or natural origin.

Statistical analysis

Statistical analyses were performed in Microsoft Excel 365 (Microsoft 2018) and R Studio 2023.03.1+446. For all indices (geo-accumulation, pollution load, potential ecological risk) at all depths, a one-sided 95% T-confidence interval was used to determine the lower 95% confidence bound. Those values were compared to reference values (TEL, PEL) for a determination of the minimal potential contamination (to 95% confidence). Differences among replicate cores were determined using matched pairs t-tests with Bonferroni-Holms correction and significance level 0.05. For statistical purposes, half of the limit of detection was used for non-detected (nd) samples. Differences in median heavy metal concentrations at different depths at each site and for each contaminant were assessed using a Kruskal-Wallace test with Wilcoxon Rank-Sum and Bonferroni-Holms correction follow-up at significance level 0.05.

Results

Heavy metal concentrations in port, control, and reef sites

All port sediment cores collected were from an area that has a significant disturbance signature due to past dredging. Concentrations of all 14 heavy metals were detected in all sediment cores and surface reef sediment (Tables S4–S17). Table 1 illustrates the geometric mean heavy metal concentrations, from greatest to least, across each sediment core while Table 2 shows all sediment concentration ranges, arithmetic means, and geometric means per location. Metal concentrations at respective depths across all cores per location revealed depth variability in most metals at only DCC and PHQ (Table S18). Due to inconsistencies of metal concentration variation by depth, data were compiled across all 10 port sediment cores and all subsamples. Likewise, the data were aggregated for all the subsamples of the two control site cores and the two reef locations. Port sediment cores have the highest geometric mean concentrations in eight of the 14 heavy metals (As, Cd, Cr, Co, Mo, Ni, Se, and V), while the control site has the highest concentrations for five heavy metals (Cu, Pb, Mn, Hg, and Zn), and the reef sediment sites have the highest for Sn. Mercury, Cd, Co, and Se were consistently the lowest concentrations across all locations—port, control, and reef. The greatest similarities in geometric mean concentrations among the three locations were found for As, Cd, Cr, and Hg.

Table 1 Order of geometric mean heavy metals concentrations from greatest to least for each port, control, and reef duplicate sediments (1, 2, 3).

Duplicate sediment locations	Order of heavy metal concentrations	
DCC1	Mn >As >Sn >Mo >V >Zn >Cu >Cr >Ni >Se >Pb >Co >Cd >Hg	
DCC2	Mn >As >Sn >V >Mo >Cr >Zn >Ni >Cu >Se >Pb >Co > Cd >Hg	
DCC3	Mn >As >Mo >Zn >V >Sn >Cr >Cu >Ni >Se >Co >Pb > Cd >Hg	
PEC1	Mn >V >Zn >Cr >Cu >As >Pb >Ni >Mo >Co >Se >Sn >Cd >Hg	
PEC2	Mn >V >Cr >Zn >As >Pb >Cu >Ni >Mo >Sn >Co >Se >Cd >Hg	
PEC3	Mn >V >Cr >Zn >As >Cu >Ni >Pb >Sn >Mo >Co >Se >Hg >Cd	
PHQ1	V >Mo >As >Mn >Cr >Zn >Ni >Cu >Sn >Pb >Se >Co >Cd >Hg	
PHQ2	Mo >As >V >Mn >Cr >Ni >Zn >Cu >Sn >Se >Pb >Co >Cd >Hg	
STB1	Mn >Zn >V >Cu >As >Cr >Mo >Ni >Pb >Sn >Se >Co >Cd >Hg	
STB2	Zn >As >V >Mn >Mo >Cu > Cr >Sn >Ni >Pb >Se >Co >Hg >Cd	
WL1	Mn >Zn >As >Pb >Cr >Cu >V >Ni >Sn >Mo >Se >Co >Hg >Cd	
WL2	Mn >Zn >As >V >Cr >Pb >Cu >Ni >Sn >Se >Mo >Co >Hg >Cd	
NR1,2,3	Mn >As >V >Cr >Zn >Sn >Pb >Cu >Ni >Se >Co >Cd >Mo >Hg	
SR1,2,3	Mn >Zn >Cr >V >As >Cu >Sn >Pb >Ni >Se >Co >Cd >Mo >Hg	
NR1	Mn >As >V >Cr >Zn >Sn >Pb >Cu > Ni >Se >Co >Cd>Mo >Hg	
NR2	Mn >Cr >V >As >Zn >Sn >Pb >Cu >Ni >Co >Se >Cd >Mo >Hg	
NR3	Mn >V >As >Cr >Zn >Sn >Pb >Cu >Ni >Se >Co >Cd >Mo >Hg	
SR1	Zn >Cu >Mn >Cr >V >As >Sn >Pb >Ni >Se >Co >Cd >Mo >Hg	
SR2	Mn >Cr >V >Zn >As >Sn >Pb >Ni >Cu >Se >Co >Cd >Mo >Hg	
SR3	Mn >Zn >Cr >V >As >Pb >Sn >Ni >Cu >Se >Co >Cd >Mo >Hg	
Notes.

DCC Dania Cutoff Canal

PEC Park Education Center

PHQ Park Headquarters

STB South Turning Basin

WL West Lake

NR North Reef

SR South Reef

Table 2 Range, arithmetic mean (mean), and geometric mean values of 14 heavy metals analyzed in Port Everglades and West Lake sediment cores and reef surface sediment samples.

Heavy metals	Port range (μg/g)	Port mean (μg/g)	Port geometric mean (μg/g)	West Lake range (μg/g)	West Lake mean (μg/g)	West Lake geometric Mean (μg/g)	Reef range (μg/g)	Reef mean (μg/g)	Reef geometric mean (μg/g)	
As	0.607–223	16.1	8.82	1.7–21.7	8.5	7.25	2.41–8.55	5.23	4.75	
Cd	nd–0.916	0.10	0.05	nd–0.282	0.0702	0.0433	0.010–0.039	0.031	0.029	
Cr	0.155–56.8	11.0	6.88	0.337–11.2	5.28	4.59	4.70–7.55	6.20	6.12	
Co	0.0238–7.40	0.542	0.325	0.119–0.520	0.270	0.257	0.030–0.100	0.048	0.044	
Cu	0.004–215	11.9	2.82	0.260–30.4	7.44	4.11	0.510–28.6	5.35	1.25	
Pb	0.0169–73.8	3.79	1.54	0.145–16.4	6.18	4.42	0.900–1.80	1.39	1.36	
Mn	1.61–204	33.0	19.7	9.38–93.8	52.0	45.4	10.1–24.1	15.3	14.8	
Hg	nd–0.736	0.0265	0.001	nd–0.262	0.0386	0.00477	nd–0.0046	0.0016	0.0013	
Mo	nd–385	17.8	4.96	nd–3.61	0.558	0.206	nd–0.040	0.0038	0.0021	
Ni	0.232–29.3	4.64	2.80	0.438–16.8	2.12	1.67	0.410–0.860	0.659	0.643	
Se	nd–4.79	0.83	0.45	0.115–0.885	0.419	0.377	0.040–0.120	0.083	0.078	
Sn	nd–140	6.78	1.25	0.064–7.70	0.757	0.538	0.990–2.07	1.62	1.57	
V	0.160–176	20.7	10.8	0.432–59.6	6.28	4.50	3.80–9.12	5.47	5.18	
Zn	0.112–603	21.7	5.27	0.409–49.8	15.7	10.2	2.76–91.1	18.7	6.80	
Notes.

Nd is non-detected. Bolded values show the highest mean and geometric mean concentrations values per location.

Molybdenum geometric mean concentrations varied widely per location, as much as four orders of magnitude. Although Mo is moderately to highly represented at all the port sites (4.96 µg/g), it is the second lowest heavy metal concentration at the RF site (0.0021 µg/g), and the third lowest concentration at the control site (0.206 µg/g).

The highest geometric mean metal concentration across all sites is observed for Mn (45.4 µg/g) at the control site followed by a close grouping of V (10.8 µg/g), Zn (10.2 µg/g), As (8.82 µg/g), and Cr (6.88 µg/g) (Table 2).

Continental crust value comparison

Port sediment samples display the highest heavy metal concentrations above continental crust values (Table S3). Of the 258 port, 34 control, and six reef sediment samples analyzed, As concentrations exceed continental crust value (1.5 µg/g) 89%, 100%, and 100% of the time, respectively. Molybdenum concentrations exceed the continental crust value (1.5 µg/g) 79%, 6%, and 0%, at the three respective locations. Tin concentrations exceed crust value (5.5 µg/g) 22%, 3%, and 17% respectively, while Cu (25 µg/g) exceed 12%, 6%, and 17% at the respective locations (Tables S4–S17).

Geo-accumulation index (Igeo)

Geo-accumulation index measures the pollution intensity of individual sample locations and is a quantitative measure of the contamination degree in sediments relative to background continental crust values (Förstner, 1980). Arsenic is the only heavy metal that exhibits strongly to extremely contaminated geo-accumulation values (4–5) in the port and control sites but moderate to no contamination (0–2) at the reef sites. All sites have at least moderately contaminated geo-accumulation values of As. Statistical analysis indicates that the average As geo-accumulation values of the duplicate cores at the DCC (1.85–5.05) and STB (1.07–2.79) sites show contamination throughout the cores, while the PEC (0–2.58) and PHQ (0–4.02) cores mostly have As contamination, with some sections having no contamination. The control WL cores, and the two RF sites show moderate As contamination with values ranging 0.74–2.50 and 0.53–1.68, respectively (Table S19).

Molybdenum also exhibits strongly to extremely contaminated geo-accumulation values (4–5) in all port cores, but shows no contamination in the control and reef sites. All sites have at least moderately contaminated geo-accumulation values of Mo. Statistical analysis indicates that the average Mo geo-accumulation values of the duplicate cores at the STB (1.91–3.71) site show moderate to strong contamination throughout the cores while the DCC (−2.73–4.71), PEC (0–2.70), and PHQ (0–4.99) cores mostly have Mo contamination with some sections having with no contamination. The control WL cores shows no overall Mo contamination (Table S19).

Pollution load index (PLI)

The ratio of metal concentration and background continental crust value yields the pollution load index. Based on the PLI, the DCC and PEC port cores have polluted surface sediments. DCC and PEC sediment cores have a PLI>1.00 for DCC1 at 5 cm, DCC2 at 20 and 25 cm, PEC2, and PEC 3 at 5 cm. No significant PLI>1.00 were found for PHQ, STB, WL, and the RF sites, indicating no pollution (Table S20).

Threshold and probable effect levels (TEL and PEL)

Threshold effect level (TEL) and a probable effect level (PEL) have been derived for nine metals (As, Cd, Cr, Cu, Hg, Ni, Pb, tributyltin, Zn) that occur in Florida coastal sediments (Macdonald, 1994). Of the 258 port, 34 control, and six reef sediment samples collected, each site has varying TEL and PEL values for As. All four port sites have As concentrations above TEL (7.24 µg/g) and PEL (41.6 µg/g). Within the port, 96% of DCC samples are above As TEL, while 38% are above PEL. The PEC site has 38% of its samples above TEL and 0.9% above PEL. PHQ has 78% of its samples above TEL and 4% above PEL, while STB has 88% of its samples above As TEL and 4% above PEL. The control site, WL, has 58% of its samples above TEL and none above PEL. The reef sites, RF, have 17% above As TEL and none above PEL (Table S21).

Copper exceeds the TEL (18.7 µg/g) in samples at all port locations except one site (PHQ). The port cores have up to 60% of samples that exceed Cu TEL, while the control site has 11%, and the reef 17%. Two port sites, PEC and STB, have 2% and 8% of samples, respectively, with Cu above PEL (108 µg/g) but the control reef sites have none.

Port cores also have up to 22% of samples exceeding Ni TEL (15.9 µg/g) while 6% of the control samples exceed TEL. All port sites (up to 27% of samples) have samples exceeding Zn TEL (124 µg/g), while only the port DCC site has samples (2%) above PEL (271 µg/g). The Zn concentrations at the control site do not exceed TEL. Mercury concentrations exceeding TEL (0.13 µg/g) are present in 14% of samples at only one port site, PEC, and in 17% of samples of the control site, WL. Reef sites do not exceed TEL for Ni, Zn, and Hg. (Tables S4–S17).

Potential ecological risk (PER)

The combined metal concentrations within each sediment sample determines the potential ecological risk (Guo, Xianbin & Zhanguang, 2010). Sediment samples are ranked within five levels, ranging from low to moderate to considerable to high to significantly high. All port sites exhibit moderate to significantly high potential ecological risk. Statistical analyses show that the average PER values for DCC (76.5–755), PHQ (12.1–427), and STB (127–368) sites range from moderately to significantly high-risk levels; PHQ has two five cm sections (5 and 10 cm) with low PER (<40). PEC (11.1–441) has moderate to high risk levels throughout the cores and significantly high (332 for PEC 3 and 441 for PEC 2) at the surface (top 5 cm). The control site, WL, exhibits moderate to considerable potential ecological risk (28.2–137) while the reef sites display low to moderate risk 44.6 and 57.7 (Table S22).

Discussion

Port Everglades has been repeatedly subjected to multiple historic major and maintenance dredging events since its establishment (USACE, 1998; PEDBC, 2000; USACE, 2005; USACE, 2020). These anthropogenic disturbances have altered the sedimentation rates and distribution through mixing and removal of surface sediment. Analysis of metal concentrations in the sediment cores at respective depths showe variability in most metals at DCC and PHQ locations (Table S18), but not PEC or STB. Both DCC and PHQ locations are more exposed and not semi-enclosed as the other STB, PEC, and control sites. However, accurate sedimentation rates and associated heavy metal concentrations could not be determined and for this reason metal concentrations at specific depths cannot be well understood nor explained. This study focuses on the overall contamination of the sediments and the possible impact they can have to the nearby benthic communities, especially with a looming large deepening and widening dredging event and its potential environmental impact (PEDBC, 2023; USACE, 2022).

Heavy metal concentrations among duplicate/triplicate cores

Port sample locations PEC, PHQ, and WL cores have no significant differences among their duplicate cores based on the geo-accumulation, pollution load, and pollution ecological risk indices. DCC cores 1, 2, and 3 show significant differences in the geo-accumulation index for Cd, while STB cores 1 and 2 show significant differences in the geo-accumulation and the ecological risk indices for Cu, Mo, Se, and Zn (Table S23). Except for the STB cores, there are little to no differences among the duplicate cores collected. Even so, it is still advisable to collect multiple cores per location for reproducibility.

Arsenic contamination in sediment

Arsenic is the contaminant that exceeds continental crust values in almost all the locations (89% port, 100% WL, and 100% RF of samples). The DCC locality has the highest concentrations of As where 96% of its samples are above TEL and 38% above PEL values at depths between 10–50 cm. The geo-accumulation index indicates strong to extreme As contamination in the port and WL, and moderate contamination at the RF. Results from port localities (PEC, PHQ, STB, and DCC) show 38–96% of the samples exceed the TEL value and 0.9–38% exceed the PEL value. The WL and RF localities exceed TEL values in 58% and 17% of their samples, respectively (Tables S4–S17).

Port sediments were expected to have higher concentrations of heavy metals due to 100 years of port activities. However, it was unexpected that the WL control site has As concentrations above TEL values, since it is distant from the port with no motorboat activity. Even so, port sediments have much higher overall As concentrations throughout their cores (Fig. 2) compared to WL (Fig. 3). WL core sediments have low heavy metal contamination except for As. The As concentrations in the port sediment (0.607–223 µg/g) are also among the highest determined in marine sediment found in ports and estuaries worldwide, except for a location in Estaque, France that ranged from 107–220 µg/g and at an English estuary that was contaminated with 1,740 µg/g with acid mine waste (Table S24).

Figure 2 Arsenic concentrations per five cm depth of one sediment core for each port location.

Dania Cutoff Canal (DCC), Park Education Center (PEC), Park Headquarters (PHQ), and South Turning Basin (STB) compared with the continental crust value of 1.5 µg/g the threshold effect level (TEL) of 7.24 µg/g and the probable effect level (PEL) of 41.6 µg/g.

Figure 3 Arsenic concentrations per 5 cm depth of the sediment cores for the control West Lake (WL) location compared with the continental crust value of 1.5 µg/g, the threshold effect level (TEL) of 7.24 µg/g, and the probable effect level (PEL) of 41.6 µg/g.

In general, As is one of the most common toxins introduced anthropogenically through herbicides, pesticides, plant defoliants, wood preservatives, cattle-dipping vats, chicken litter, coal burning, trash incineration, water treatment sludge, and industrial sources (Macdonald et al., 1996; NRC, 2000; Rosen & Liu, 2009; Missimer et al., 2018). The As found in both the port and WL locations could be the result of runoff and atmospheric deposition. Between 1992–1993, As concentrations (13–51 µg/g) were detected in reclaimed water at Bonaventure Golf Course, located due west of Port Everglades along the L-35 canal, which flows to the Intracoastal Waterway and Port Everglades (Fig. 4). The As originates from the monosodium methanearsonate that was applied to the turf as an herbicide to control grasses and weeds (Swancar, 1996). Arsenic can also be released during municipal solid waste incineration and deposited back to the environment (Hu et al., 2015). During the high temperature incineration process, arsenic is predominantly present as As2O3(g) vapor and as a result, concentrations are usually elevated in local marine environments and bioaccumulate through trophic webs (Volesky, 1990; Hu et al., 2015).

Figure 4 Map of Broward County Florida depicting the canals, Intracoastal Waterway, and Port Everglades.

Map was created using ArcGIS® software (esri) and Google Earth. Data for the canals was provided by the Hydrography, Drainage Areas, and STAs open data on the South Florida Waste Management District database.

One of the major sinks for As from municipal waste incineration is in sediments (Pierce & Moore, 1982). Broward County’s Environmental Monitoring Laboratory (BCEML) reported concentrations of As (0.623–19.4 µg/g) in marine surface sediment in the C-11, C-12, C-13 canals, and Intracoastal Waterway from 2005–2007 (Bernhard, 2014). In 2005, Bernhard (2014) reported As concentrations below 2.43 µg/g in surface marine sediment from multiple Intracoastal Waterway locations north of Port Everglades to the middle branch of New River. The results from this project indicate As concentrations that are up to ten times higher at certain depths in the port sediment (0.607–223 µg/g) while the mean value of 16.1 µg/g is in line with the higher concentrations of As determined compared to the BCEML data. The control site had approximately equal As concentrations (1.7–21.7 µg/g) and mean of 8.5 µg/g, while the reef site As concentrations were lower (2.41−8.55 µg/g) with a mean of 5.23 µg/g compared to the BCEML data (Table 2). The port and control site sediment cores were taken at variable depths, while the BCEML data were for surface sediment only, which could explain the differences in concentrations. This study indicates the importance of collecting sediment cores from the port since concentrations vary per depth and dredging events take sediment from variable depths.

In its inorganic forms, As is lethal to organisms and harmful to the environment (Jaishankar et al., 2014). Its toxicity to marine organisms is complex due to the existence of two inorganic arsenic species As (III), as arsenite, and As (V), as arsenate; both can be found in aquatic ecosystems (Liber, Doig & White-Sobey, 2011). Arsenite is more lipid soluble than arsenate and has the highest acute toxicity among the different As forms (Saha et al., 1999). Studies with freshwater invertebrates determined that arsenite is generally more toxic than arsenate (Borgmann, 1980; Spehar et al., 1980; Golding, Timperley & Evans, 1997). Various sublethal effects on behavior, growth, locomotion, reproduction, and respiration indicate that As is acutely toxic to marine organisms (Macdonald et al., 1996).

Molybdenum contamination in sediment

Molybdenum concentrations exceed continental crust values in 79% of all port sediment samples, only 6% in WL, and none in RF. Its concentrations are higher in the port sediment (Fig. 5) than WL (Fig. 6) and RF by as much as four orders of magnitude. The geo-accumulation index indicates strong to extreme Mo contamination in port sediment, low to moderate in WL, and none in RF. The Mo concentrations in the port sediment (nd-385 µg/g) are amongst the highest determined in marine sediment found in ports and estuaries worldwide, where values ranged 0.5–40 µg/g (Table S24).

Figure 5 Molybdenum concentrations per 5 cm depth of one sediment core for each port location.

Dania Cutoff Canal (DCC), Park Education Center (PEC), Park Headquarters (PHQ), and South Turning Basin (STB) compared with the continental crust value of 1.5 µg/g.

Figure 6 Molybdenum concentrations per five cm depth of the sediment cores for the control West Lake (WL) location compared with the continental crust value of 1.5 µg/g.

In higher oxic conditions, Mo accumulates at concentrations close to the crustal concentration abundance (1.5 µg/g), while in anoxic and sulfidic conditions Mo is enriched (Crusius et al., 1996). The differences observed among the three locations for Mo may result from more oxic conditions found at the RF sites. Molybdenum concentrations in sediment, along with Fe distribution, is used to identify oxic conditions in marine systems (Scholz et al., 2017). Molybdenum can be transferred from the water column to sediments through particulate shuttle effects involving Fe and Mn and has been found in areas bearing manganese oxides (Crusius et al., 1996; Algeo & Lyons, 2006; Algeo & Tribovillard, 2009). To our knowledge, Mo concentrations and its possible effects have not been reported for Broward County, Florida.

Manganese contamination in sediment

Manganese has the overall highest geometric mean concentration (45.4 µg/g) of all the heavy metals analyzed (Table 2). Although Mn occurs naturally in sediments and soils due to the weathering of parent material, anthropogenic processes such as mining, smelting and addition of biosolids/organic wastes to agricultural areas increase Mn concentrations (Paschke, Valdecantos & Redente, 2005; Boudissa et al., 2006; Li et al., 2013). Concentration ranges of heavy metals in surface waters at six sampling sites near Port Everglades between 1975–1978 indicated Mn concentrations of nd-120 µg/L (Sonntag, 1980). In addition, in 1983 heavy metal concentrations of lithophilic samples at varying depths at the Dixie water-treatment plant site adjacent to US Highway 441, Broward County, were <1–26 µg/kg (Howie & Waller, 1986). In comparison, the sediment results from the port, control, and reef sites in this study indicate much higher Mn concentrations. The port samples have 1.61–204 µg/g (mean: 33.0 µg/g), the control 9.38–93.8 µg/g (mean: 52.0 µg/g), and the reef sites 10.1–24.1 µg/g (mean: 15.3 µg/g) (Table 2). These higher concentrations could be due to Mn sediment accumulation over time through runoff.

There is little data on how Mn affects the biota in marine environments and TEL and PEL values have not been established for Mn; however, tropical species appear to be more sensitive to Mn concentrations than temperate ones (Anastasi & Wilson, 2010). Even though Mn concentrations did not exceed the background continental crust value of 600 µg/g, the overall concentrations can be of concern to benthic organisms since studies indicate sensitivity of benthic organisms to certain Mn concentrations. Summer, Reichelt-Brushett & Howe (2019) showed reduced hard coral fertilization success after exposure to 164–237 µg/g of Mn, larval death at 7 mg/L, and adult coral tissue death at 0.7 mg/L (where mg/L is equivalent to µg/g). In our study, the concentration of Mn in the port sediment is 1.61–204 µg/g, 9.38–93.8 µg/g at the control site, and 10.1–24.1 µg/g at the reef. The proximity of the coral reef tract to the port warrants concern should Mn in the port sediment remobilize during dredging. All port and control site cores were collected along mangrove channels, and Mn has been associated with mangrove forest rhizospheres and high permeability of sediments (Guieros et al., 2003). Studies have shown that seagrass uptakes heavy metals (Smith et al., 2019) and mangroves contribute to heavy metal retention (Nguyen, Le & Richter, 2020). In the marine environment, aquatic plants play a crucial role in the uptake, storage, and recycling of heavy metals (Chandra & Kulshreshtha, 2004; Smith et al., 2019). The Mn concentrations in the port sediment (1.61–204 µg/g) are equivalent or even slightly below marine sediment found in ports and estuaries worldwide (0.4–4643 µg/g) (Table S24).

Copper contamination in sediment

Copper concentrations exceed continental crust values in 12% of all port sediment samples, 6% in WL, and 17% in RF samples. As many as 60% of port sediment samples exceed TEL values, and 11% of WL samples and 17% of RF samples exceed TEL values (Tables S4–S17).

Copper is a common metal that exists in three forms, Cu, Cu (I), and Cu (II). Its ionic forms are biologically available since it is essential for the proper growth of animals (Flemming & Trevors, 1989; Ikemoto et al., 2004). Weathering of copper-bearing metals, copper sulfides, and copper are natural sources of this element within marine ecosystems (Macdonald et al., 1996). Anthropogenic sources of copper include copper wire mills, coal burning, smelting, refining, and iron and steel producing industries (Health Canada, 2019). Copper is an essential micronutrient and is therefore easily and readily accumulated by marine organisms and plants (Macdonald et al., 1996). However, Cu can be toxic to bivalve mollusks, altering the biochemical and physical properties of the surface epithelium and disrupting membrane permeability (Cheng, 1979). Dean, Shimmield & Black (2007) discovered that a significant amount of Cu in marine sediment at fish farms originated from anti-fouling agents. These can be released in either solid or particulate form originating from traps, coated nets, or hard structures (i.e., container ships) into the marine environment (Claisse & Alzieu, 1993; Miller, 1994; Brooks, 2000; Morrisey et al., 2000; Solberg, Sæthre & Julshamn, 2002; Brooks & Mahnken, 2003).

Concentration ranges of Cu in surface waters at six sampling sites near Port Everglades between 1975–1978 were nd-55 µg/L (Sonntag, 1980). In the 1980s Cu concentrations in stormwater runoff from residential, commercial, and roadway lands in Broward County, Florida were nd-500 µg/L (Whalen & Cullum, 1988). Copper concentrations in highway runoff, shallow groundwater, and lithophilic samples in 1983, at varying depths at the Dixie water-treatment plant site adjacent to US Highway 441, Broward County, had values ranging from <10–100 µg/L, <10–40 µg/L, and <1–10 µg/kg, respectively (Howie & Waller, 1986). BCEML reported the concentration of Cu (3.3–413 µg/g) in marine surface sediment in the C-11, C-12, C-13 canals, and intracoastal waterway from 2005–2007 (Bernhard, 2014). In 2005, Bernhard (2014) reported Cu concentrations (6.05–153 µg/g) in surface marine sediment from multiple Intracoastal Waterway locations north of Port Everglades to the middle branch of New River. The results from this study indicate Cu accumulation in the port, control, and reef sites sediment similar to the 2005–2007 and 2014 studies, with the port values ranging from 0.004–215 µg/g (mean: 11.9 µg/g), the control from 0.260–30.4 µg/g (mean: 7.44 µg/g), and the reef sites from 0.510–28.6 µg/g (mean: 5.35 µg/g) (Table 2). The Cu concentrations in the port sediment are also equivalent or slightly below marine sediment found in ports and estuaries worldwide (0.06–1195 µg/g), except for a location at an English estuary that was contaminated with 2,398 µg/g from acid mine waste (Table S24).

Contaminant sources

South Florida consists of vast stretches of undeveloped wetlands, such as the Everglades, agricultural/horticultural lands (row crop, citrus, pasture) and municipal developments such as roadways, man-made canals, golf courses, commercial, and residential structures. For over a century, canals, locks, dams, and levees that were created along the Everglades altered as much as 70% of the natural water flow. So instead of waters flowing southward to Florida Bay and the Gulf of Mexico, it has diverted to estuaries on the Gulf and Atlantic coasts (Fig. 4; Haman & Svendsen, 2006).

Stormwater

It is common that coastal environments are contaminated by urban stormwater, agricultural runoff, and domestic and industrial wastewater. Stormwater runoff and associated contaminants are of particular concern in Florida due to its high annual precipitation. Urban stormwater’s most common pollutants are nutrients (typically from agriculture), sediments, and metals. Due to the substantial population growth in recent years, as well as the proximity of urban developments to the Florida coast, urban stormwater is a major source of contaminants (Castro et al., 2013).

Broward County analyzed trends in heavy metal concentrations of surface waters (equal proportion of fresh and marine water samples), relative to rainfall over a 27-year period (Broward County DNR, 1998). Surface water from 1970–1997 were measured for nine heavy metal (As, Cd, Cr, Cu, Fe, Ni, Pb, Sn, and Zn) concentrations, which comprised the majority of our studied 14 metals. Broward County compared the concentrations of metals detected in their 1996–1997 study with those reported in the New River Basin study in 1991–1992 and those taken from EPA’s storage and retrieval (STORET) national database from 1970 to 1996. In the Broward County 1996–1997 study, the highest levels of metals in surface waters coincided with rainfall, with 44% of all metals detected occurring during a heavy rain day. The highest concentrations (µg/L) recorded in the study were 20.1 for As, 17.1 for Cd, 27.2 for Cr, 25.2 for Cu, 2470 for Fe, 13.5 for Ni, 277 for Pb, 68.6 for Sn and 304 for Zn. Data from the 1970–1996 EPA STORET for Broward County had higher values with As, Cu, Pb, and Zn concentrations (µg/L) reaching 200, 90, 400, and 510, respectively. Although the heavy metal concentrations in the stormwater runoff are lower than the concentrations in the sediment from this study, high rainfall activity in South Florida can lead to high accumulation of heavy metals in the sediment over time.

Fertilizers, animal waste, and pesticides

Substantial quantities of fertilizers and pesticides are used in Florida to increase agricultural product yields. Therefore, poorly managed runoff from agricultural areas and collection ponds into the Everglades severely affects receiving water systems (Zhiwei, 2012; Castro et al., 2013). Heavy metals and other contaminants transported through agricultural surface runoff leads to metal toxicity and ecological risks (Alengebawy et al., 2021).

The management of animal waste products from industrial feeding operations, where a high population of animals are in a confined area, has become a priority in Florida (Stade, 2018). Concentrated animal feeding operations (CAFOs) use heavy metals such as Cu and Zn as supplements to help promote growth and prevent disease in pigs and chickens (Jacela et al., 2010). Other metals noted in animal waste include Cd, Pb, Hg, and As (Daia et al., 2016). The waste created from the animals is then sprayed on farm fields where these metals accumulate in the soil and can contaminate water supplies through runoff, which can flow to the Atlantic coast estuaries (Jackson et al., 2003; Burkholder et al., 2007; Rodríguez, McLaughlin & Pennock, 2018).

Even though there is little commercial agriculture and horticulture in Broward County, major freshwater runoff to southeast Florida comes from Lake Okeechobee which passes through the major agricultural and horticultural hub of the state. Runoff could subsequently transport heavy metal contaminants associated with fertilizers, pesticides, and animal waste.

Atmospheric deposition

The porous geology and high-water table exhibited throughout Florida creates a pathway for heavy metal contamination to reach the groundwater. Ash from coal-fired power plants is deposited in landfills where toxic heavy metals readily leach into water (SCAF, 2022). Groundwater wells along the west coast of Florida near a coal-fired power plant tested as high as 79.8 µg/L of As, 502 µg/L of Li, and 338 µg/L of Mo (Geosyntec Consultants, 2019). The US Environmental Protection Agency has set maximum groundwater contaminant levels for As, Li and Mo to 10, 40 and 100 µg/L, respectively (USEPA, 0000a).

Additionally, the combustion of municipal waste generates fly ash, contributing to ecosystem contaminants through atmospheric deposition and water transport. Fly ash contains 5–30% dioxins, furans, polyaromatic hydrocarbons (PAHs), and metals such as As, Cd, Cr, Ni, and Pb (Chrostowski & Sager, 1990). South Florida has two superfund sites near Port Everglades that were used to combust municipal waste. The Wingate Road Municipal Incinerator Dump (EPA identifier: FLD000807156) is located five miles northwest of Port Everglades along the New River, a direct drainage into the ICW and Port Everglades. The city operated the waste incineration facility from 1954–1978. Then in 1989 the EPA placed the site on the Superfund program’s National Priorities List (NPL) because of contaminated soil, sediment, and surface water (USEPA, 2016). From 2013–2015, sediment from Rock Pit Lake located within the site had As concentrations from 0.65–59 mg/kg and Cd concentrations from 1.4–260 mg/kg. Groundwater from the same site between 2011–2014 had As concentrations from 3.3–5 µg/L. This is indicative of sediment accumulating As over time since the ground water concentrations were much lower, but the lake sediments have similar As concentrations to the marine sediments in this study. The Davie Landfill (EPA Identifier: FLD980602288) is in southern Florida approximately 14 miles west southwest of Port Everglades along a canal that drains into the port. The waste management facility was operated by Broward County from 1964–1987 (USEPA, 0000b). In 1983 the EPA placed the site on the Superfund program’s National Priorities List (NPL) due to contaminated groundwater, sludge, and soil. In the 1970s, samples from the New South River Canal (C-11 Canal) along the landfill boundary exhibited levels of Cr, As, and Pb that prompted health concerns (USEPA, 0000b).

Ecological impacts to benthic communities

The port sediment heavy metal concentrations have the potential to ecologically impact benthic organisms. Port Everglades, a commercial port, is intermittently dredged to accommodate the draft of large ships. The dredging of this port can present a contaminant threat to benthic organisms and the adjacent coral reef community, as most sediment collected in the port displays evidence of high heavy metal contamination and ecological risk.

The resuspension of contaminated dredged marine sediment and the remobilization of heavy metals can have adverse ecological effects on resident aquatic organisms. Wei et al. (2021) determined that dredged materials at the dumping area of Huangmao Island, China had a negative impact on the benthic community and that larger dumping amounts lead to higher heavy metal pollution risks. Kalnejais, Martin & Bothner (2010) determined that sediment resuspension was responsible for transferring significant amounts of metals (Fe, Mn, Ag, Cu, and Pb) to the mobile dissolved phase in Boston Harbor, Massachusett. Fetters et al. (2016) revealed that redeposited suspended sediment increased the bioavailability and toxicity of Zn, Cu, Cd, Pb, Ni, and Cr to marine Hyalella azteca (amphipod crustacean). Banks et al. (2012) investigated the effect of trace metals (including As) during an extended oxygen depletion event across the sediment-water interface in sediments from a highly metal contaminated estuary in southeast Tasmania, Australia. Where hypoxia developed, Mn and Fe were released from sediments at all sites, while As, Cd, Cu and Zn release was comparatively low. Ferrans et al. (2021) assessed the metal pollution risk during present and future dredging at Malmfjärden Bay, Sweden. Contamination and ecological risk factors indicated low Cr, Ni and Fe pollution concern and a medium risk for Pb and Zn during dredging activities.

More research could help to understand how dredging and other types of disturbances on marine geochemistry affect the resuspension and release/remobilization of sediment contaminants, such as heavy metals. Even so, it is known that the disposal of dredged material is the largest mass input of waste deposited into the oceans (Vivian & Murray, 2009).

The use of ecological indices is a common and accepted practice in assessing sediment heavy metal contamination and risks to the respective environments. However there are limitations using ecological indices such as geo-accumulation, pollution load, and potential ecological risk. These indices use sediment metal concentration data and synthesize it in an understandable way, but it can also be overly generalized. A single index alone does not provide the whole story of ecological risks and/or contamination. An unusually high reading at one location or depth could skew the index outcome. This is why multiple ecological indices are used to provide a more comprehensive assessment of contamination and help translate the heavy metal data into understandable measures of ecological risks.

Conclusions

Overall, heavy metal concentrations in Port Everglades sediment have similar values to other anthropogenically altered coastal locations, including ports and estuaries. Arsenic and Mo concentrations are consistently higher in Port Everglades than those found in other global port and estuary locations, while Cd, Cr, Co, Hg, Pb, Mn, and Ni have lower mean concentrations. Copper, Se, Sn, V, and Zn in the port sediments have comparable concentrations to those found in other studies. Arsenic is the only heavy metal to have slightly elevated concentrations at the control site. Based on the results of this study, overall heavy metal contamination, as well as As and Mo contamination, is distinct in Port Everglades sediment, while both the control and reef sites have moderate to little contamination. Currently, there are no heavy metal ecological risks associated with the surface sediment at the adjacent coral reef sites, but that condition could change with future dredging operations. The port sediments have the potential to harm the benthic communities through heavy metal remobilization, especially during large dredging projects as is projected for Port Everglades. More research could help to assess the connection between remobilization of contaminated sediment and its impact on the marine environment.

Supplemental Information

Supplemental Information 1 Split sediment cores for Dania Cutoff Canal (DCC), Park Education Center (PEC), Park Headquarters (PHQ), South Turning Basin (STB), and West Lake (WL)

The left side of the core at 0 cm is the top (surface). Munsell sediment color chart is also shown.

Click here for additional data file.

Supplemental Information 2 Inductively coupled plasma mass spectrometer (ICPMS) detection limits (µg/g) for the 14 heavy metals tested

Click here for additional data file.

Supplemental Information 3 Recovery % for standard reference material, SRM 2702-inorganics in marine sediment to evaluate reliability of analytical method

Click here for additional data file.

Supplemental Information 4 Threshold Effect Level (TEL), Probable Effect Level (PEL), and continental crust values

NA is not available.

Click here for additional data file.

Supplemental Information 5 Dania Cut-off Canal 1 heavy metal concentrations (µg/g) by sediment core depth (cm), minimum (min), maximum (max), median, arithmetic mean (mean), and geometric mean (geomean)

N/a = end of sediment core. N/d = Not detected. For statistical purposes half of the limit of detection was used for n/d samples. Bold indicate maximum concentration values.

Click here for additional data file.

Supplemental Information 6 Dania Cut-off Canal 2 heavy metal concentrations (µg/g) by sediment core depth (cm), minimum (min), maximum (max), median, arithmetic mean (mean), and geometric mean (geomean)

N/a = end of sediment core. N/d = Not detected. For statistical purposes half of the limit of detection was used for n/d samples. Bold indicate maximum concentration values.

Click here for additional data file.

Supplemental Information 7 Dania Cut-off Canal 3 heavy metal concentrations (µg/g) by sediment core depth (cm), minimum (min), maximum (max), median, arithmetic mean (mean), and geometric mean (geomean)

N/a = end of sediment core. N/d = Not detected. For statistical purposes half of the limit of detection was used for n/d samples. Bold indicate maximum concentration values.

Click here for additional data file.

Supplemental Information 8 Park Education Center 1 heavy metal concentrations (µg/g) by sediment core depth (cm), minimum (min), maximum (max), median, arithmetic mean (mean), and geometric mean (geomean)

N/a = end of sediment core. N/d = Not detected. For statistical purposes half of the limit of detection was used for n/d samples. Bold indicate maximum concentration values.

Click here for additional data file.

Supplemental Information 9 Park Education Center 2 heavy metal concentrations (µg/g) by sediment core depth (cm), minimum (min), maximum (max), median, arithmetic mean (mean), and geometric mean (geomean)

N/a = end of sediment core. N/d = Not detected. For statistical purposes half of the limit of detection was used for n/d samples. Bold indicate maximum concentration values.

Click here for additional data file.

Supplemental Information 10 Park Education Center 3 heavy metal concentrations (µg/g) by sediment core depth (cm), minimum (min), maximum (max), median, arithmetic mean (mean), and geometric mean (geomean)

N/a = end of sediment core. N/d = Not detected. For statistical purposes half of the limit of detection was used for n/d samples. Bold indicate maximum concentration values.

Click here for additional data file.

Supplemental Information 11 Park Headquarters 1 heavy metal concentrations (µg/g) by sediment core depth (cm), minimum (min), maximum (max), median, arithmetic mean (mean), and geometric mean (geomean)

N/a = end of sediment core. N/d = Not detected. For statistical purposes half of the limit of detection was used for n/d samples. Bold indicate maximum concentration values.

Click here for additional data file.

Supplemental Information 12 Park Headquarters 2 heavy metal concentrations (µg/g) by sediment core depth (cm), minimum (min), maximum (max), median, arithmetic mean (mean), and geometric mean (geomean)

N/a = end of sediment core. N/d = Not detected. For statistical purposes half of the limit of detection was used for n/d samples. Bold indicate maximum concentration values.

Click here for additional data file.

Supplemental Information 13 South Turning Basin 1 heavy metal concentrations (µg/g) by sediment core depth (cm), minimum (min), maximum (max), median, arithmetic mean (mean), and geometric mean (geomean)

N/a = end of sediment core. N/d = Not detected. For statistical purposes half of the limit of detection was used for n/d samples. Bold indicate maximum concentration values.

Click here for additional data file.

Supplemental Information 14 South Turning Basin 2 heavy metal concentrations (µg/g) by sediment core depth (cm), minimum (min), maximum (max), median, arithmetic mean (mean), and geometric mean (geomean)

N/a = end of sediment core. N/d = Not detected. For statistical purposes half of the limit of detection was used for n/d samples. Bold indicate maximum concentration values.

Click here for additional data file.

Supplemental Information 15 West Lake 1 heavy metal concentrations (µg/g) by sediment core depth (cm), minimum (min), maximum (max), median, arithmetic mean (mean), and geometric mean (geomean)

N/a = end of sediment core. N/d = Not detected. For statistical purposes half of the limit of detection was used for n/d samples. Bold indicate maximum concentration values.

Click here for additional data file.

Supplemental Information 16 West Lake 2 heavy metal concentrations (µg/g) by sediment core depth (cm), minimum (min), maximum (max), median, arithmetic mean (mean), and geometric mean (geomean)

N/a = end of sediment core. N/d = Not detected. For statistical purposes half of the limit of detection was used for n/d samples. Bold indicate maximum concentration values.

Click here for additional data file.

Supplemental Information 17 North Reef (NR) heavy metal concentrations (µg/g) of surface sediment samples (5 cm) with minimum (min), maximum (max), median, arithmetic mean (mean), and geometric mean (geomean)

N/d = Not detected. For statistical purposes half of the limit of detection was used for n/d samples.

Click here for additional data file.

Supplemental Information 18 South Reef (SR) heavy metal concentrations (µg/g) of surface sediment samples (5 cm) with minimum (min), maximum (max), median, arithmetic mean (mean), and geometric mean (geomean)

N/d = Not detected. For statistical purposes half of the limit of detection was used for n/d samples.

Click here for additional data file.

Supplemental Information 19 Metal concentration significant variations across depths for all cores per location. X denotes significant Kruskal-Wallace test results. No Wilcoxon pairwise with Holm correction follow-ups were significant

Click here for additional data file.

Supplemental Information 20 Geo-accumulation indices and statistical analyses for Mo and As of all the cores per location and depth

Bolded numbers indicate some degree of contamination. Avg = average; StErr = standard error; CI LB = confidence interval lower bound; 4 - 5: strongly to extremely contaminated; 3 - 4: strongly contaminated; 2 - 3: moderately to strongly contaminated; 1 - 2: moderately contaminated; 0 - 1: uncontaminated to moderately contaminated; < 0: uncontaminated.

Click here for additional data file.

Supplemental Information 21 Pollution load indices (PLI) and statistical analyses for all cores and sediments per depth

Bolded numbers (PLI < 1) indicate pollution is present. Avg = average; StErr = standard error; CI LB = confidence interval lower bound.

Click here for additional data file.

Supplemental Information 22 Arsenic concentrations (µg/g) and statistical analyses for all cores and sediments per depth

Orange values are above threshold effect level (TEL 7.24 µg/g) and red above probable effect level (PEL 41.6 µg/g). Avg = average; StErr = standard error; CI LB = confidence interval lower bound.

Click here for additional data file.

Supplemental Information 23 Potential ecological risk (PER) and statistical analyses for all cores and sediments per depth

PER > 320 significantly high ecological risk (dark red), 160 < PER < 320 high ecological risk (red), 80 < PER < 160 considerable ecological risk (orange), 40 < PER < 80 moderate ecological risk (yellow), PER < 40 low ecological risk. Avg = average; StErr = standard error; CI LB = confidence interval lower bound.

Click here for additional data file.

Supplemental Information 24 P-Value, Matched-Pairs T-Test with B-H Correction statistics depicting differences between duplicate cores per location when comparing to geo-accumulation and pollution ecological risk indices

Sites that only had significant differences are shown

Click here for additional data file.

Supplemental Information 25 Marine sediment heavy metal concentration ranges in Port Everglades, FL, worldwide ports and estuaries

* port; # location with contaminated acid mine waste; n/d = non-detected.

Click here for additional data file.

Any use of trade, firm, or product names is for descriptive purposes only and does not imply endorsement by the U.S. Government. We thank Dr. Alan Shiller and Melissa Gilbert at the University of Southern Mississippi’s Center for Trace Analysis for providing the ICPMS analyses.

Additional Information and Declarations

Competing Interests

Author Contributions

Field Study Permissions

Data Availability

The authors declare there are no competing interests.

Dimitrios G. Giarikos conceived and designed the experiments, performed the experiments, analyzed the data, prepared figures and/or tables, authored or reviewed drafts of the article, and approved the final draft.

Laura White conceived and designed the experiments, performed the experiments, analyzed the data, prepared figures and/or tables, authored or reviewed drafts of the article, and approved the final draft.

Andre M. Daniels conceived and designed the experiments, performed the experiments, authored or reviewed drafts of the article, and approved the final draft.

Radleigh G. Santos analyzed the data, prepared figures and/or tables, authored or reviewed drafts of the article, and approved the final draft.

Paul E. Baldauf conceived and designed the experiments, performed the experiments, prepared figures and/or tables, authored or reviewed drafts of the article, and approved the final draft.

Amy C. Hirons conceived and designed the experiments, performed the experiments, analyzed the data, authored or reviewed drafts of the article, and approved the final draft.

The following information was supplied relating to field study approvals (i.e., approving body and any reference numbers):

Permits for sediment core collection were provided by Broward County, U.S. Army Corps of Engineers, and Florida Department of Environmental Protection.

The following information was supplied regarding data availability:

The data is now available at NSUWorks: Giarikos, Dimitri G., ”Assessing the Ecological Risk of Heavy Metal Sediment Contamination from Port Everglades Florida” (2020). SECLER Data. 1.

https://nsuworks.nova.edu/secler_data/1.

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
