# Peer review of "Assessing the ecological risk of heavy metal sediment contamination from Port Everglades Florida USA"

_PeerJ, doi:10.7717/peerj.16152_

## Round 0.1 · original submission · Major Revisions

The reviewers were generally pleased with the article. However, there were some important comments made; please address all of the comments and resubmit.

Reviewer 1 ·

Basic reporting

Basic reporting
The article discusses the assessment guidelines for addressing concerns related to coastal ecosystem contamination, specifically in Florida's coastal sediments. The study derived numerical SQAGs for nine metals that commonly occur in coastal sediments, namely As, Cd, Cr, Cu, Hg, Ni, Pb, tributyltin, and Zn. The TEL and PEL were developed as powerful tools to assess the contaminant levels in sediment, with the former being the concentration at which a toxic response is observed in benthic organisms, while the latter is the concentration at which a large percentage of benthic organisms exhibit a toxic response. The potential ecological risk index (PER) was also calculated to determine the degree of contamination for combined metal concentrations within each sediment sample.

Statistical analysis was performed to determine the heavy metal concentrations in port, control, and reef sites. The results showed that concentrations of all 14 heavy metals were detected in all sediment cores and surface reef sediment. Port sediment cores had the highest geometric mean concentrations in eight of the 14 heavy metals, while the control site had the highest concentrations for five heavy metals, and the reef sediment sites had the highest for one heavy metal. Mercury, Cd, Co, and Se were consistently found to have the lowest concentrations across all locations. The highest geometric mean metal concentration across all sites was observed for Mn at the control site. The article also compares the heavy metal concentrations to the continental crust value and discusses the Geo-accumulation index, which measures the pollution intensity of individual sample locations and is a quantitative measure of the contamination degree in sediments relative to background continental crust values.

Overall, the study provides a comprehensive assessment of coastal ecosystem contamination concerns in Florida's coastal sediments. The use of SQAGs, TEL, PEL, PER, and statistical analysis provides a powerful tool for assessing the levels of contaminants in sediment, and the results demonstrate the varying concentrations of heavy metals in port, control, and reef sites. The article also highlights the importance of comparing heavy metal concentrations to the continental crust value and the Geo-accumulation index for a more comprehensive understanding of the pollution intensity in individual sample locations. These findings can be used as a reference for future studies on coastal ecosystem contamination and aid in developing effective mitigation strategies

Experimental design

Experimental design and Validating
Review comments : Technical evaluation
1. The methods used for subsampling, washing, drying, and digestion of sediment samples are described in detail, which is essential for ensuring the reproducibility and accuracy of the results. However, there are some potential sources of variability that could affect the quality of the data. For instance, the use of ultrapure deionized water may not completely remove all the contaminants from the sediment samples, leading to an underestimation of the metal concentrations. Additionally, the use of different drying ovens may result in different moisture contents of the sediments, which could affect the weight measurements and subsequent metal concentrations. It would be helpful if the authors discussed these potential sources of error and their impact on the data.

2. The heavy metal analysis method used in this study is standard and widely used, which adds to the validity of the results. However, the authors should provide more information on the accuracy and precision of the ICP-MS method used for measuring the metal concentrations. For example, it would be useful to know the detection limits, recoveries, and precision of the method. The use of a standard reference material is commendable, but the authors should provide more information on how the results compare to the certified values of the reference material.

3. The comparison of the heavy metal concentrations in the sediment samples to a control site is a reasonable approach to assess the extent of metal pollution in the study area. However, the authors should provide more information on how the control site was chosen and why it is considered representative of a background level. In addition, the use of continental crust values as a reference for background levels is questionable, as the composition of marine sediments is likely to be different from that of continental crust. The authors should consider using a more appropriate reference material, such as deep-sea sediments or pristine marine sediments.

4. The use of the geo-accumulation index and pollution load index to assess the degree of contamination in the sediment samples is a valid approach. However, the authors should provide more information on how the contamination factors were calculated and how they relate to the metal concentrations. In addition, the use of threshold effect levels and probable effect levels would be useful to determine the ecological risk of the metal pollution. Overall, the methods used in this study are reasonable, but there are some limitations and potential sources of error that should be discussed in more detail.

Validity of the findings

5. The methods used for analyzing the sediment samples appear to be thorough and rigorous. The use of the US EPA digestion method 3050B for sample digestion is a widely accepted standard method for the analysis of heavy metals in sediments. The heavy metal analysis using a sector-field inductively coupled plasma mass spectrometer is a highly sensitive and accurate method that allows for the detection of low levels of heavy metals in the sediment samples. The comparison of heavy metal concentrations in the sediment samples to the background elemental composition concentrations of the Earth's upper crust is a useful tool for assessing the degree of contamination.

6. The use of the geo-accumulation index and pollution load index to quantify the degree of contamination in the sediment samples is a standard method in environmental science. However, it is important to note that the use of these indices does not necessarily provide a complete picture of the environmental impact of heavy metal contamination. Other factors such as the bioavailability of heavy metals and their effects on local ecosystems must also be considered.

7. Overall, the methods used for analyzing the sediment samples appear to be appropriate for the research question being addressed. The use of the geo-accumulation index and pollution load index provides a useful tool for assessing the degree of heavy metal contamination in the sediment samples. However, it is important to consider the limitations of these indices and the potential for other factors to affect the environmental impact of heavy metal contamination. Can you elaborate more on this.

Additional comments

General / Non-technical comments
1. The degree of contamination was evaluated using the geo-accumulation index and the pollution load index, and threshold effect levels and probable effect levels were determined by the Florida Department of Environmental Protection. Overall, the methods used appear thorough and detailed, allowing for a comprehensive analysis of the sediment samples.

Overall review
Based on the information provided in the paragraph, it appears to contain a thorough description of the methodology used in the study, including details on the sampling, processing, and analysis of sediment samples. However, to improve the manuscript's chances of acceptance, it may be helpful to clarify the research questions or hypotheses being tested, provide more context on the significance of the findings, and ensure that the writing is clear, concise, and follows the journal's guidelines for formatting and style.

Reviewer 2 ·

Basic reporting

This manuscript reports on the distribution of various trace metals at different sites in the Port Everglades, FL, USA. The manuscript links the need to understand the metals' concentration in port with the dredging activities carried out at the port to increase its bathymetry, and the potential impact of such action on the adjacent coral reef system close to the port. Sampling was carried out at selected locations in the port, a nearby "control" area and the coral reef. The manuscript is well-written, provides valuable data on the distribution of trace metals along the sediment cores, from seabed surface to 2 m depth, with high vertical resolution. Although it reads like a technical report, I believe it merits publication in PeerJ provided that major review is made in the original submission.

Experimental design

The experimental design to collect sediment cores, maintain samples, collect sub-samples from each core, perform lab analysis and compute the geochemical indices is valid and according to the technical standards followed in similar studies worldwide. Methodology is described in adequate detail by the authors. My main comments on the Materials and Methods section are:
a) How do you define a "control location"? Is this area unaffected by any human activities? Will you use these metal levels as background levels in indices estimation? Why not use the bottom sample of your port sediment cores as control?
b) Authors collect sediment cores from locations having depth less than 2 m. How representative of the port environment and sediments are these cores? Is the water body margin representative of the main port environment?
c) Authors did not examine the chronology of each split core. This is something that could give added value to your work and link the metals' concentration to the various human activities taking place at each respective period.
d) Sediment grain size analysis was not performed, which is a drawback for such studies.

Validity of the findings

Results are presented in tables, comparing metals content in the study area with that found in similar cases from literature. However, these comparisons are not adequately discussed. Moreover, it is unclear how authors processed these results. Did they aggregate results for all port cores and subsamples? Changes found along each split core at respective depths are not discussed. ANOVA analysis would help results interpretation.

Additional comments

All additional comments are given in the attached annotated manuscript.

Annotated reviews are not available for download in order to protect the identity of reviewers who chose to remain anonymous.

Reviewer 3 ·

Basic reporting

Since As is not classified as a metalloid element and was the major contaminant in this study, I would recommend using another word instead of "heavy metal".

There are only a few typing mistakes, for example, at line 175 on the equation.

Figures 2, 3, 4, and 5:
Would it be better to put the average values (possibly the geometric mean) of the core taken from the same area with their error bar?
For continental crust values, TELs, and PELs, I would recommend using a straight line without markers when they are the reference values (not from the measurement).

Abstract: This work is high in value. However, in the abstract, only your results are shown. It would be better to add a few more sentences, according to your discussion or the implication of your research.

The rest is excellent.

Experimental design

Excellent, well done

Validity of the findings

+ According to the figures presented;
-Interestingly, at around 125–140 cm of Mo (Fig. 4), PEC and PHQ showed peaks. Please explain the possibility of these peaks. [also any As at the same depth]
+It would be wonderful if you had some information about the sedimentation rate in this area that would help us understand the previous possible time when the contaminant deposited.

---

## Round 0.2 · accepted · Accept

Thank you for taking the time to address the reviewers' comments. I am satisfied that they are addressed sufficiently. There were a number of comments that suggested that the reviewers did not understand that the sediment cores you collected were from an area that has a significant disturbance signature due to past dredging. Perhaps some text in the abstract or at the beginning of the results section reminding readers of this will avoid confusion.